# Preparation of Xylan-*g*-/P(AA-*co*-AM)/GO Nanocomposite Hydrogel and its Adsorption for Heavy Metal Ions

**DOI:** 10.3390/polym11040621

**Published:** 2019-04-04

**Authors:** Weiqing Kong, Minmin Chang, Chunhui Zhang, Xinxin Liu, Bei He, Junli Ren

**Affiliations:** State Key Laboratory of Pulp and Paper Engineering, South China University of Technology, Guangzhou 510640, China; kongweiqing1119@gmail.com (W.K.); 17728109456@163.com (M.C.); lxx19910312@163.com (X.L.); hb2453102334@163.com (B.H.); renjunli@scut.edu.cn (J.R.)

**Keywords:** xylan, graphene oxide, hydrogel, adsorption, heavy metal ions

## Abstract

Xylan-*g*-/P(AA-*co*-AM)/Graphene oxide (GO) hydrogels were prepared and used in the removal of heavy mental ions. Acrylamide (AM), acrylic acid (AA), and xylan were used as the raw materials to prepare the hydrogels with ammonium persulfate (APS) as the initiator. The prepared hydrogels were characterized by Fourier transform infrared (FTIR), thermogravimetric analysis (TGA), and energy dispersive X-ray (EDX). Some important properties of nanocomposite hydrogels such as swelling behavior, mechanical property, and adsorption capacity were also examined as well as the regeneration of the hydrogels. The results showed that the prepared hydrogels reached the equilibrium state of swelling after 12 h, and the compressive strength of the hydrogel with 30 mg of GO could reach up to 203 kPa. Compared with traditional hydrogel, the mechanical properties of the hydrogels with GO were obviously improved. The maximum adsorption capacity of hydrogels for Pb^2+^, Cd^2+^, and Zn^2+^ could reach up to 683 mg/g, 281 mg/g, and 135 mg/g, respectively. After five cycles of adsorption and desorption, the recovery rate of the hydrogels on Pb^2+^, Cd^2+^, and Zn^2+^ was still up to 87%, 80%, and 80%, respectively—all above 80%.

## 1. Introduction

The problems of ecological environment have become increasingly prominent, prompting people to start looking for pollution control measures. In the 21st century, heavy metal pollution of water environment has become a worldwide concern. Even small amounts of heavy metals can have a significant impact on the health of people. There are many methods that can be used to remove metal ions such as chemical precipitation, ion exchange, chemical oxidation/reduction, and electrodialysis [1]. However, these methods have their own shortcomings, especially with regard to their high costs [2]. Adsorption is an effective method for the removal of heavy metal ions. Among them, bio-adsorption has become a potential technology in removing heavy metal ions [3]. More attention has been focused on the exploration of renewable biomass as the raw materials for preparation of bio-adsorbents including starch, cellulose, lignin, chitin/chitosan, and agricultural wastes [4,5,6,7,8].

Hemicellulose, as the renewable polymer, is biocompatible, reproducible, and easily functionalized. It has broad application prospects in the fields of chemistry, materials, and biology [9,10]. Hemicellulose includes lots of hydroxyl and some carboxyl, which have a remarkable ability to bind heavy metal compounds [11]. The adsorption capacity of hemicellulose itself is not high, therefore, more adsorption groups were introduced to enhance the adsorption effect. Xylan is a kind of hemicellulose with a backbone of β-(1→4) linked D-xylosyl residues substituted with other saccharide units [12]. It is the main polysaccharide of the cell walls of hardwoods that accounts for 20–35 wt% of the biomass [1]. Abundant groups on xylan could be further reacted with other polymers to form new materials [13,14,15]. Xylan-based materials have been studied and used as a kind of adsorbent for the removal of pollutants such as heavy metal ions and methylene blue [1,14].

Recently, many studies have proved that hydrogel possessed great adsorption capacity and reusability [16,17,18]. Hydrogel contains a three-dimensional network, and a number of active groups in the polymer can react with heavy metal ions easily. The pore of hydrogel increases the surface area and also improves adsorption performance. Xylan-type hemicellulose can form unstable hydrogel, which can be dissolved by the chemical modification of xylan. A few studies have paid attention to the preparation of xylan-based hydrogel. In one study for example, the adsorbent consisting of xylan-rich hemicelluloses was prepared by the polymerization of acrylic acid (AA) and the hemicelluloses [1]. The prepared adsorbent was used to remove the heavy metal ions including Pd^2+^, Cd^2+^, and Zn^2+^. The results showed that the hydrogel exhibited great adsorptive properties and great regeneration. It can be reused without obvious loss of adsorption capacity for Pd^2+^,6 Cd^2+^, and Zn^2+^ after a lot of adsorption/desorption cycles. Adsorption capacities of the adsorbent for Pd^2+^, Cd^2+^, and Zn^2+^ could reach to 859, 495, and 274 mg/g, respectively. A kind of hydrogel based on hemicellulose was prepared via graft co-polymerization with AA, and the hydrogel was used for the removal of Pd^2+^ [19]. The results showed a maximum mono-layer adsorption capacity of 5.88 mg/g. Adsorbents could become a very promising direction for the utilization of hemicellulose. However, after swelling, the xylan-based hydrogel became brittle and difficult to recycle due to the low mechanism property. These drawbacks could be overcome by improving the strength of xylan-based hydrogels, which could in turn broaden the application of xylan-based hydrogel.

Graphene oxide (GO) is known as functional modified graphene, and is one of the great materials because of its good surface activity [20]. Its main structure is similar to graphene, except that it contains a large number of oxygen-containing groups, mainly including –OH, –C=O, –COOH and –C–O–C. These active groups cause GO to have good hydrophilicity, meaning it is easy to disperse in water. Although the insertion of oxygen-containing groups breaks the tight arrangement of carbon atoms in graphene to some extent—making GO less conductive than graphene—the stable structure of GO enables the strong mechanical properties of graphene to be preserved. A large number of active groups provide a large number of reaction sites for the modification of GO and form a strong interface interaction between GO and the polymer during the composite process, which is widely used to improve the polymer properties.

In this work, xylan was used as the raw material to prepare hydrogel for the removal of heavy metal ions. Graphene oxide (GO) was used as the reinforcing filler, and *N*, *N*-methylenebisacrylamide (MBA) as the crosslinking agent. As monomers, acrylic acid (AA) and acrylamide (AM) were also used to prepare the hydrogels through copolymerization and crosslinking. The mechanical properties of hydrogels were evaluated. Also, the adsorption performance of prepared hydrogels was discussed, such as GO, pH, and adsorption time. The adsorption kinetics and adsorption–desorption were also studied.

## 2. Materials and Methods

### 2.1. Materials

Bagasse xylan (*M*_w_ of 49,000 g·mol^−1^, xylose > 85%) was supplied by Yuanye (Shanghai, China). High purity flake graphite was provided by Nanjing Xianfeng Nano Technology Co., LTD (Nanjing, China). AA, AM, APS, and MBA were obtained from Fuchen Chemical Reagent Factory (Tianjin, China). *N*,*N*,*N*,*N*-tetramethylethylenediamine (TEMED) was purchased from MACKLIN (Shanghai, China). NaOH and absolute ethyl alcohol were supplied by Jinhuada Chemical Reagent Factory (Guangzhou, China). N_2_ was purchased from Wang Qi Trade Co. LTD (Guangzhou, China). HNO_3_, Pb(NO_3_)_2_, Cd(NO_3_)_2_, and Zn(NO_3_)_2_ were purchased from Tianjin Kemiou Chemical Reagent Co. LTD (Tianjin, China).

### 2.2. Preparation of GO

GO was synthesized using a modified Hummers method [21]. First, 1 g of high-purity flake graphite and 0.5 g of sodium nitrate were added to 46 mL of concentrated sulfuric acid and stirred in an ice bath. Then, 3.2 g of KMnO_4_ was added slowly while stirring for 1.5 h under 35 °C. 46 mL of deionized water was mixed and stirred for 0.5 h under 98 °C. After the reaction was over, the reactor was cooled down to room temperature and 140 mL of deionized water and 10 mL of 30% hydrogen peroxide solution were added until the mixture turned bright yellow. A hydrochloric acid solution (1:10) was used to remove the metal ions. Then the resulting products were centrifuged with deionized water until the pH value was about 6. Finally, the product was freeze-dried to obtain GO powder.

### 2.3. Preparation of Xylan-g-P(AA-co-AM)/GO Hydrogels

First, 1 g of xylan was dissolved in 1 mL ultra-pure water at 85 °C for 1.5 h until xylan was completely dissolved. Under the condition of an ice bath, 10 mL of the GO solution (2 mg/mL) was added to the xylan solution and bubbled with nitrogen for 15 min. After the addition of 3 g of AA and 2 g of AM, 0.05 g of APS, 0.036 g of MBA, and 40 uL of promoter were also added. When the solution was completely homogeneous under stirring, this mixed solution was transferred to a small beaker and reacted in 60 °C for 2 h. The obtained hydrogel was washed with excess distilled water to remove the residual reaction reagent in the hydrogel and then freeze dried.

### 2.4. Characterization of Prepared Hydrogels

FTIR spectra were recorded in KBr disks using a spectrum instrument (TENSOR27, Bruker, Karlsruhe, Germany). The spectrum was obtained in the range of 4000–400 cm^−1^. Energy dispersive X-ray (EDX) was used to examine the surface content of hydrogels at an acceleration voltage of 10 kV (EVO 18, Carl Zeiss, Jena, Germany). The compression performance of hydrogels was measured using a universal testing machine (INSTRON 5565, Instron Corporation, Boston, Massachusetts, USA). The XRD of GO was conducted using polycrystal X-ray diffraction (Rigaku, SmartLab SE, Tokyo, Japan). The stability of the hydrogels was detected by thermal gravimetric analyzer (TA Q500, TA Instruments, Newcastle, DE, USA). The TEM of GO was tested by the transmission electron microscope (JEM-2100F, JEOL, Tokyo, Japan). The hole area of the hydrogels was recorded by the high-performance automatic mercury injection instrument (AutoPore IV 9500, Micromeritics, Norcross, GA, USA). The swelling behavior of hydrogels was evaluated by the gravimetric method. First, 0.1 g of dried hydrogels was recorded, then hydrogel was immersed in ultra-pure water at room temperature. After a while, the swollen hydrogel was removed from water, dried with clean filter paper, and weighed. The swelling ratio (SR) was calculated using Equation (1):SR = (W_t_ − W_d_)/W_d_(1)
where W_t_ was the weight of the swollen hydrogel and W_d_ was the initial weight of the dried hydrogel.

### 2.5. Metal Ion Adsorption and Desorption of Hydrogels

Pb^2+^, Cd^2+^, and Zn^2+^ were used as the heavy metal ions to study the adsorption of hydrogels. Pb(NO_3_)_2_, CdCl_2_·2.5H_2_O, and Zn(NO_3_)_2_·6H_2_O were dissolved in deionized water with concentrations of 50–800 mg/L. A certain amount of 0.1 mol/L HNO_3_ and NaOH solutions were prepared to regulate the pH. Ten milligrams of dry hydrogels were weighed and added into 50 mL of Pb^2+^, Cd^2+^, and Zn^2+^ solutions with a concentration of 200 mg/L. The pH was adjusted in the range of 4.5–6.5 using 0.1 mol/L of HNO_3_ and NaOH solution. The mixture was placed in a constant temperature oscillator for 6 h at room temperature. The metal ion concentration was determined by atomic absorption spectrometer. The adsorption capacity of hydrogels was calculated using Equation (2):q_e_ = (C_0_ − C_e_)V/m(2)
where q_e_ was the adsorption capacity (mg/g) of hydrogels, C_0_ and C_e_ were the initial concentration and adsorption equilibrium concentration (mg/L) of the metal ion solution, V was the volume of the used metal ion solution (L), and m was the mass of dry hydrogel (g).

After adsorption, hydrogels were immersed in HNO_3_ solution for 2 h and the solution was retained. Hydrogels were soaked in 1 mol/L NaOH solution and washed by deionized water until the pH value reached 7.0. The obtained hydrogels were placed in an oven at 50 °C to dry the constant weight again for adsorption. The adsorption–desorption process was repeated 5 times. The desorption capacity of hydrogels to metal ions can be calculated by Equation (2).

## 3. Results and Discussion

### 3.1. Characterization of Hydrogels

The xylan chain contains many hydroxyl groups, which could react with ammonium persulfate (APS) to produce a large number of free radicals, which in turn could initiate the polymerization reaction of acrylic acid (AA) and acrylamide (AM) on the branch chain of xylan. Due to the presence of GO sheets, the polymerization also occurs between GO sheets. In this process, the GO contains active functional groups such as hydroxyl (–OH) and carboxyl (–COOH), which could also form covalent bonds or hydrogen bonds through indirect branching polymerization of polymer chains. Under the action of the crosslinking agent MBA, xylan/GO/P(AA-*co*-AM) hydrogel forms a complex network, which is displayed in Scheme 1.

Figure 1 shows the TEM and XRD of GO. As seen in the spectrum of XRD, GO has the characteristic peak at 2 θ = 11.5, which proved the nano size between the interplanar of GO [22]. In TEM, GO has the very thin layer, and the folds are because of the oxygen-containing groups [23].

Figure 2 shows the FTIR spectra of prepared hydrogels. Obviously, 3419, 2916, 1623, 1460, 1421, 1384, 1323, 1251, 1164, 1114, 1080, 1043, 981, and 894 cm^−1^ are all characteristic absorption peaks of xylan [24]. Among them, the absorption peak at 3419 cm^−1^ is from the stretching vibration of hydroxyl group on xylan. The absorption peaks at 1384 and 1251cm^−1^ are derived from the bending vibration of C-H and the stretching vibration of C-O. The absorption peak at 1043 cm^−1^ is related to the glycosidic bonds in the xylan hemicellulose molecular chain. Compared with the raw xylan hemicellulose, prepared hydrogels produce several new absorption peaks in the infrared spectrum. The absorption peaks at 1658, 1564, and 1452 cm^−1^ are all related to the symmetric stretching vibration of C=O, which proves that the hydrogel contained –COO– from monomer acrylic acid and GO [25]. The stretch vibration and bending vibration of N–H at 1658 cm^−1^ are also related to the graft polymerization monomer acrylamide [26]. The absorption peak of hemicellulose at 3419 cm^−1^ shifted to 3431 cm^−1^ and the absorption peak became wider, which was caused by the intermolecular hydrogen bond formed between the hydroxyl group on the hemicellulose molecular chain and the active group on GO and N–H.

Figure 3 illustrates the thermogravimetric analysis (TGA) curves of the hydrogels with or without GO. As shown in the figure, there is no obvious difference among them and every curve has the same trend. When the temperature was elevated from room temperature to 700 °C, weight losses of the samples included four stages: 25−220 °C, 220−350 °C, 350−400 °C, 400−700 °C. The weight loss in the first stage was because of the evaporation of water and the degradation of small molecules. The weight loss at 220−400 °C was due to the degradation of the long-chain molecules such as xylan, polyacrylamide, and polyacrylic acid. Then, in the final stage, the weight of the hydrogels remained constant, which was attributed to the carbonation of the hydrogels. As seen in the figure, the hydrogels with higher content of GO had the higher weight in the end, which indicated that GO has a positive effect on the thermostability of hydrogels. The addition of GO also enhanced the intermolecular forces.

### 3.2. Swelling Behavior Studies of Hydrogels in Deionized Water

The swelling capacity of hydrogels is very important to their application. Figure 4 shows the swelling kinetics curve of hydrogels with or without GO. Swelling behavior of hydrogels is a process of dynamic equilibrium. Water molecules could easily diffuse into the hydrogel to expand its volume; besides, the expansion of the hydrogel makes the molecular chain of the network generate the elastic shrinkage. When these two opposite tendencies are opposed to each other, a dynamic equilibrium of swelling is reached. Clearly, the swelling rate of hydrogels increased rapidly in the initial stage. When the swelling time was 12 h, the swelling rate hardly changed. Therefore, the hydrogel reached the equilibrium state of swelling in the water after 12 h.

In addition, the swelling rate of the hydrogel with GO and the equilibrium swelling were higher than the hydrogel without GO. This is because the insertion of GO sheets increased the pore density of the hydrogel, and the large number of hydrophilic oxygen-containing groups of GO improved the density of hydrophilic oxygen-containing groups of the hydrogel. The swelling rate also increased accordingly.

### 3.3. The Effect of GO on the Mechanical Properties of Hydrogels

The compression performance of the hydrogel before and after the addition of GO was tested with the universal testing machine. Importantly, GO had an obvious positive influence on the mechanical properties of hydrogels. As shown in Table 1, the compression strength increased from 11.5 kPa to 95 kPa after the addition of GO. The addition of GO effectively improved the mechanical properties of the hydrogel. The GO structure made xylan macromolecules interwoven with each other. Polymerization of AA and AM between the layers of GO also occurred. The polar groups on the side chain of P(AA-*co*-AM) form a hydrogen bond with the oxygen-containing functional group on the surface of GO. AA and AM was also grafted onto xylan chains to form a dense hydrogel, which enhanced the mechanical properties of the hydrogel.

### 3.4. Metal Ion Adsorption of Hydrogels

#### 3.4.1. FTIR and EDX of the Hydrogels after Adsorption

Figure 5 shows the FTIR spectra of the hydrogels after adsorbing heavy metal ions. As shown in Figure 2, the peaks at 1654 cm^−1^ and 1562 cm^−1^ are related to the symmetric stretching vibration of C=O, which proves the presence of –COO^−^. However, after adsorbing the Pb^2+^, the absorption peaks of –COO^−^ are shifted to 1649 cm^−1^ and 1541 cm^−1^, respectively. After adsorption of Cd^2+^, the absorption peaks of –COO^−^ are shifted to 1662 cm^−1^ and 1552 cm^−1^, respectively. After adsorption of Zn^2+^, the absorption peaks of –COO^−^ are shifted to 1710 cm^−1^ and 1654 cm^−1^, respectively. These changes are due to the formation of –COO^−^ compounding with Pb^2+^, Cd^2+^, and Zn^2+^ in the hydrogel network during the adsorption process, which proves the chelation effect between –COO^−^ and heavy metal ions. An EDX was used to examine the surface content of the hydrogels after adsorbing the three kinds of heavy metal ions. As Figure 6 shows, the surface of the hydrogels contains heavy metal ions.

#### 3.4.2. The Influence of GO on the Adsorption Properties of Hydrogel

GO was applied in this work as the reinforcer for improving strength of hydrogels. However, as a carbon-based material, GO has a large number of oxygen-containing functional groups such as –OH, –COOH, and C=O, which makes it become a kind of adsorbent [27]. In the Figure 7, after the addition of GO, the adsorption capacity of the xylan-*g*-/P(AA-*co*-AM)/GO hydrogels increased for the three kinds of metal ions, especially for the adsorption capacity of Pb^2+^ because the high content of –OH and –COOH in GO provided more ligands for adsorbing metal ions, which improved the adsorption capacity of metal ions.

Figure 8 showed the hole area of the hydrogels without or with 20 mg and 30 mg of GO. As seen in the figure, the hole area increased with the increase of the addition of GO. The hydrogels with more GO had the bigger adsorption capacity according to the Figure 7 and Figure 8, which might be related to the oxygen-containing groups. The hydrogels with a bigger hole area exposed more oxygen-containing groups, which increased the adsorption capacity of the hydrogels.

#### 3.4.3. The Influence of pH on the Adsorption Properties of Hydrogel

Figure 9 illustrates the influence of pH on the adsorption property of hydrogels. When the pH was 3.0, the adsorption capacity of hydrogels on Pb^2+^, Cd^2+^ and Zn^2+^ was low. With the increase of pH, the adsorption capacity of hydrogels on metal ions increased. This is due to the protonation of active functional groups in the hydrogel network at lower pH values, which hinders the interaction between adsorbents and heavy metal cations [1]. When the pH value is lower, the protonation of the amine group (–NH_2_) in the hydrogel was more obvious. There is electrostatic repulsion between –NH_3_^+^ and divalent metal ions (Pb^2+^, Cd^2+^, and Zn^2+^), which reduces the binding sites of metal ions that can be adsorbed, resulting in reduced adsorption capacity of the hydrogel. With the increase of pH value, the protonation of amines decreased gradually, and more amines chelated with metal ions, thereby the adsorption of hydrogel increased gradually [28]. In addition, the –COOH group in the hydrogel could be changed into –COO^−^ in the range of high pH value, which generates electrostatic attraction with metal ions, and could also lead to the increase of adsorption capacity [29].

#### 3.4.4. Adsorption Kinetics of Hydrogels on Metal Ions (Pb^2+^, Cd^2+^, Zn^2+^)

The adsorption kinetics of hydrogels for metal ions were studied, which could be displayed in Figure 8. The adsorption capacity of metal ions including Pb^2+^, Cd^2+^, and Zn^2+^ increased rapidly with the increase in time, and the adsorption rate was relatively high within 0–1 h. Then, the rate gradually decreased, and the three kinds of metal ions reached the adsorption equilibrium at 120 min respectively. The adsorption process could be roughly divided into two stages. In the first stage, the swelling of the hydrogel and the electrostatic repulsion generated by a large number of –COO^−^ in the hydrogel provided a large number of channels for metal ions to enter the hydrogel. In the second stage, due to the continuous diffusion of metal ions, the carboxyl adsorption sites were reduced, and the electrostatic shielding effect was enhanced, so the network structure of the hydrogel began to shrink. Thus, the diffusion rate of metal ions into the adsorption materials was decreased.

### 3.5. The Desorption and Regeneration Properties of Prepared Hydrogels

The experimental results of adsorption and regeneration of hydrogels for Pb^2+^, Cd^2+^, and Zn^2+^ were shown in Table 2. After five cycles of adsorption–desorption, hydrogels still maintained a high adsorption capacity for metal ions, which were 580 mg/g, 224 mg/g, and 115 mg/g, respectively. Compared with the first adsorption and the fifth adsorption, the adsorption capacity of hydrogel on Pb^2+^ decreased from 660 mg/g to 580 mg/g, a reduction of 12.1%, and the recovery rate decreased from 99% to 87%. The adsorption capacity of Cd^2+^ decreased from 264 mg/g to 224 mg/g, and the recovery rate decreased from 94% to 80%. The adsorption capacity of Zn^2+^ decreased from 138 mg/g to 115 mg/g, a decrease of 16.7%, and the recovery rate decreased from 96% to 80%. It can be concluded from the above results that after five cycles of adsorption–desorption, the regenerated hydrogels still had a certain adsorption performance for metal ions, and prepared hydrogels had a good reuse performance. After 5 cycles, hydrogels still had good desorption ability and the recovery of metal ions was above 80%. Therefore, hydrogels were very suitable for the removal of heavy metal ions from water. In the desorption process, HNO_3_ generated H^+^, which could replace the metal ions of –COO^−^ bound in hydrogels to finish the desorption of metal ions. This proves that the adsorption process of hydrogels on metal ions was conducted through ion exchange and was a chemical adsorption process.

## 4. Conclusions

In this work, new xylan based nanocomposite hydrogels were obtained by copolymerization and crosslinking, using GO as the reinforcer. The addition of GO improved the properties of hydrogels such as the ability of swelling, mechanical properties, and the adsorption capacity for metal ions. The compression strength of hydrogels also increased from 11.5 kPa (without GO) to 95 kPa (with GO), indicating GO played an important role in improving the compression strength. With the increase of pH value, the adsorption capacity of hydrogels on metal ions increased gradually. When the initial concentrations of Pb^2+^, Cd^2+^, and Zn^2+^ aqueous solutions were 200 mg/L, respectively, the adsorption of prepared hydrogels for the three metal ions was balanced within 120 min. The maximum adsorption capacity for Pb^2+^, Cd^2+^, and Zn^2+^ could reach 683 mg/g, 341 mg/g, and 142 mg/g, respectively. Moreover, after five cycles, hydrogels still had good desorption ability and the recovery of metal ions was above 80%, which displayed good reuse performance.

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
