# Peer review of "Preparation of Xylan-g-/P(AA-co-AM)/GO Nanocomposite Hydrogel and its Adsorption for Heavy Metal Ions"

_polymers, 2019, doi:10.3390/polym11040621_

Round 1

Reviewer 1 Report

The authors prepare a complex hydrogel composite consisting of xylan-g/copolymer (P(AA-co-AM)/GO) using GO as a reinforcing agent. The authors explore the efficacy of this hydrogel composite on adsorption for heavy metal ions. This study does not seem to have a conceptual mistake and the material characterization can give explicit evidence on the authors’ hypotheses. However, the reviewer thinks that this study lacks several experiments and therefore presents a few major issues that may require an intensive revision. Some comments and suggestions are given below:

1. The reviewer is confused with the composition of the hydrogel. The authors claim xylan as a new absorbent, while the absorbent function of xylan is not discussed in the result (with and without xylan in the hydrogel). If the authors think amine group (-NH2) is an issue at low pH to reduce the adsorption capacity of the hydrogel, why do the authors use AM and AA to form the copolymer. There are a few alternatives that may be able to replace the AM, or even just use PAA.    

2. The authors claim that the adsorption mechanism of hydrogels for various metal ion includes electrostatic effect, chelation effect, and ion exchange effect. However, no related evidence and data are given in the result.

3. Does the porosity play a role in the adsorption capacity of the hydrogel? If yes, the data regarding the specific surface area should be provided.

4. The authors should adjust the drawing and data indicated in Scheme 1 and Figure 1 to make them big enough to read.

Author Response

We thank the reviewer for the valuable comments. And we have revise the manuscript according to the review. The response is placed in the word below. 

Reviewer 2 Report

In this manuscript, the authors prepared and characterized xylan-g-/P(AA-co-AM)/GO nanocomposite hydrogels. Several properties of these hydrogel were also investigated including swelling behavior, mechanical properties, metal ion adsorption capacity and their regeneration properties.

The paper is well-organized and the results are interesting. However, the data in this paper is not adequate and cannot fully support the conclusion. Therefore I would like to recommend this paper to published after carefully major revisions.

Suggestion on how to improve this paper are given below:

The author characterized and compare the swelling properties of hydrogels with and without GO nanocomposite. How about the water uptake? Is there any differences for the hydrogels with and without GO?

The paper provided and compared  the mechanical properties of the hydrogels with and without GO. From the results, the author thought the addition of GO can enhance the mechanical properties of hydrogel. However, the number of samples are not adequate to  support this conclusion. Please prepare and characterized a series of hydrogel with different component of GO. Moreover, what's the range of error for the compressive strength?

In figure 3, the paper provide the adsorption capacity of hydrogel for several metal ions. Please add the data (number) of adsorption in the figure 3. Moreover, compare the hydrogel with and without GO is not enough. Please provide and measure a series of hydrogel with different ratio of GO.

Author Response

(The authors gave the same response as above.)

Reviewer 3 Report

After reading the manuscript entitled:

"Preparation of xylan-g-/P(AA-co-AM)/GO nanocomposite hydrogel and its  adsorption for heavy metal ions"

I regret to say it is not suitable for publication in the current shape due to the following reasons:

1- Testing the manuscript by ithenticate software showed 43% with references and 36% without references which is considered as plagiarism.

2- The prepared samples are poorly characterized, the following techniques are necessary:

- XRD or FESEM: to proof of formation of nano composite.

- EDX: to give information about the surface content which is supposed to be responsible for adsorption.

- TEM or STEM along with DLS: to give evidence of nano size.

-TGA : give valuable information about the stability of the material.

3- It is very important for comparison purpose to compare between the modified hydrogel with the traditional hydrogel for adsorption of heavy metals. In figure 2 and 3 comparisons between two samples was performed while in figure 4, adsorption test was performed with one sample only.

Author Response

(The authors gave the same response as above.)

Round 2

Reviewer 1 Report

The authors have addressed all the required issues and I would like to recommend this manuscript to be accepted.

Reviewer 2 Report

The revised manuscript well addressed all kinds of question and suggestion. Now, I would like to recommend this paper to be published at present version.

Reviewer 3 Report

Accept